# Propagation-adaptive 4K computer-generated holography using physics-constrained spatial and Fourier neural operator

Ninghe Liu[1], Kexuan Liu[2], Yixin Yang[2], Yifan Peng [3] & Liangcai Cao [1,2] ✉

Computer-generated holography (CGH) offers a promising method to create true-to-life reconstructions of objects. While recent advances in deep learning-based CGH algorithms have significantly improved the tradeoff between algorithm runtime and image quality, most existing models are restricted to a fixed propagation distance, limiting their adaptability in practical applications. Here, we present a deep learning-based algorithmic CGH solver that achieves propagation-adaptive CGH synthesis using a spatial and Fourier neural operator (SFO-solver). Grounded in two physical insights of optical diffraction, specifically its global information flow and the circular symmetry, SFO-solver encodes both target intensity and propagation distance as network inputs with enhanced physical interpretability. The method enables high-speed 4 K CGH synthesis at 0.16 seconds per frame, delivering an average PSNR of 39.25 dB across a 30 mm depth range. We experimentally demonstrate various-depth 2D holographic projection and an adjustable multi-plane 3D display without requiring hardware modifications. SFO-solver showcases significant improvements in the flexibility of deep learning-based CGH synthesis and provides a scalable foundation to fulfill broader user-oriented requirements such as dynamic refocusing and interactive holographic display.

Computer-generated holography (CGH) enables photorealistic projection via numerical diffraction calculation[1], and has become a prominent method in various fields, including holographic display[2–7], laser fabrication[8–10], and metasurface design[11–13]. Unlike traditional optical holograms that are usually static and recorded in photographic films, CGHs can be directly uploaded to refreshable spatial light modulators (SLMs) for dynamic modulation. Since current commercial SLMs support only amplitude or phase holograms[14], CGH algorithms must satisfy corresponding constraints through mathematical optimization. Specifically, phase-only CGH (PO-CGH) has been taken as the primary focus due to its higher diffraction efficiency and broader applicability[15,16]. Conventional approaches to address the phase-only constraint include

direct encoding and iterative optimization. Direct encoding methods, such as double phase-amplitude encoding (DPAC), approximate the complex-valued light field with a phase-only field on the SLM plane[17,18]. Despite their fast computation, these encoding methods typically suffer from degraded image quality and reduced optical efficiency in experiments. Iterative methods use non-convex optimization strategies such as alternative projection[19,20] and gradient descent-based algorithms[21–23] to compute the optimal SLM pattern. However, generating high-quality CGHs often requires multiple iterations, leading to an inherent trade-off between algorithm runtime and image quality.

To overcome these challenges, deep learning has emerged as a promising approach for fast and high-quality CGH synthesis. The

[1]Weiyang College, Tsinghua University, Beijing 100084, China. [2]Department of Precision Instrument, Tsinghua University, Beijing 100084, China. [3]Department of Electrical and Electronic Engineering, The University of Hong Kong, Hong Kong SAR, China. ✉e-mail: clc@tsinghua.edu.cn

learning-based CGH algorithms, whether using supervised learning[2,24–26] or self-supervised strategies[23,27–29], aim to directly map the intensity distribution to the PO-CGH that reconstructs the desired optical field at a specified distance from the SLM plane. Enabled by mature computing infrastructures like the Graphics Processing Unit (GPU) and the powerful fitting ability of deep neural networks, the learning-based methods can render high-fidelity CGHs in real time, effectively resolving the speed-quality trade-off. However, this progress comes with an important limitation: existing CGH synthesis networks are typically constrained to a fixed optical configuration. Specifically, the propagation distance and the optical path length of the reconstructed objects are predefined before training the CGH network. When the optical setup changes, for example, if the user wants to move the reconstruction plane or stretch the object along the optical axis, the network must be retrained, which undermines the applicability of CGHs in broader user-specific scenarios. This limitation arises because current CGH networks are only able to learn the mapping relationship between target images and PO-CGHs under a fixed physical model. For the networks trained with supervised learning, all training data pairs have been collected under the same configuration. Similarly, in diffraction-driven self-supervised approaches, the propagation distance remains constant throughout training. This motivates the development of a more flexible CGH model that generalizes across varying propagation distances.

Here, we present a propagation-adaptive CGH synthesis framework, termed SFO-solver, which introduces a new dimension of flexibility to deep learning-based CGH algorithms. SFO-solver jointly incorporates the target intensity and propagation distance as inputs, serving as a generalized solver for the inverse problem of CGH optimization. The framework is grounded in two fundamental physical insights of optical diffraction: the flow of optical information and the circular symmetry of the diffraction filter. First, the integration of learnable Fourier filtering and spatial convolution equips the SFO-solver with both local sensitivity and global receptive field, which aligns closely with the information flow in optical diffraction. Second, inspired by the rotational symmetry of the angular spectrum (AS) diffraction model, the input propagation distance is encoded via a neural radiance field and mapped to a circular Fourier filter in SFO-solver's intermediate layer. This physics-constrained Fourier filtering exploits the geometric symmetry of diffraction to ensure model robustness, performance consistency, and physical interpretability. SFO-solver enables accurate PO-CGH synthesis, achieving an average peak signal-to-noise ratio (PSNR) of 39.25 dB and structural similarity (SSIM) of 0.986 across a 30 mm distance range. Leveraging its ability of continuous depth control, SFO-solver further facilitates multi-plane 3D holographic display in experimental demonstrations. The primary contribution of this work is that SFO-solver expands the learning of CGH networks to a new dimension by employing a physically interpretable neural encoding method. Such flexibility opens up possibilities for diverse user-defined applications without the need for hardware modification.

## Results

### Learning to optimize with multi-dimensional inputs

The synthesis of a PO-CGH poses a classical inverse problem, which seeks to determine the phase pattern $\Phi(x, y)$ on the SLM plane such that, after propagating certain distance $z$, the resulting optical field has an intensity profile equal to the target distribution $I(x, y, z)$. We denote this phase-only pattern as $\Phi(x, y; z)$, representing the hologram that reconstructs the desired image at distance $z$. The optimization problem can be expressed as

$$\text{argmin}_{\Phi} \left\Vert \left\vert Prop_z(\Phi(x, y; z)) \right\vert^2 - I(x, y, z) \right\Vert_2^2, \quad s.t. |\Phi| = 1 \quad (1)$$

Here, $Prop_z$ denotes the optical propagation operator, typically modeled using the AS solution of the Helmholtz equation[30,31]. Solving Eq. (1) with a neural network fits naturally within the paradigm of Learn to Optimize[32–35], where deep neural networks are employed to solve constrained optimization problems with significantly higher speed and efficiency compared to conventional iterative algorithms.

With the goal of developing a flexible neural solver for Eq. (1), it is essential to rigorously incorporate the physical process of optical diffraction in the network structure. Specifically, the model must accurately capture and encode how variations in target intensity and propagation distance influence the resulting phase solution within its latent representation. In SFO-solver, two physical insights are embedded into the learning framework: the optical information flow and the geometric symmetry of the AS diffraction filter (see "Physics-constrained encoding using Spatial and Fourier neural Operator"). The proof-of-concept demonstration is conducted as shown in Fig. 1a. Given a target intensity and a specified propagation distance, the corresponding PO-CGH is computed in real time and uploaded to the SLM. The reconstructed intensity profiles at various depths are then captured by a camera translated along the optical axis. SFO-solver supports continuous depth control within a 30 mm range (85–115 mm), which is sufficient to deliver perceptible defocus cues in holographic display applications using a 4K (3,840 × 2,160 pixels) SLM.

### Physics-constrained encoding using Spatial and Fourier neural Operator

Figure 1b illustrates the flow of information in both optical diffraction and SFO-solver's digital processing. Specifically, the optical point spread function (PSF) describes how information from a single SLM pixel flows to the reconstruction plane during diffraction. In the typical experimental setup, where the 4 K SLM (HOLOEYE GAEA-2) has a pixel pitch of 3.74 μm and the reconstruction plane is positioned approximately 10 cm away, the diffraction PSF spreads across the entire image (see Supplementary Note 1). Consequently, modifying a single pixel in the PO-CGH affects all pixels in the reconstructed output. To reverse this process computationally, the network should possess a global receptive field, allowing each pixel in the target intensity to flow back and influence any pixel in the predicted hologram. Achieving this with a traditional convolutional neural network (CNN) would require hundreds of layers. Instead, we integrate a Fourier neural operator[36–39] with spatial convolution to form the network backbone, as depicted in Fig. 1c. The Fourier branch provides a global receptive field through a single 2D Fourier transform, while the spatial branch captures local details with convolution layers. A learnable multi-channel filter is applied in the frequency domain and optimized via gradient descent. In this way, SFO-solver learns to optimize Eq. (1) jointly in the spatial and Fourier domains.

In addition to spatial encoding, SFO-solver incorporates the propagation distance through a physics-constrained frequency representation. In the AS formulation, the propagation distance $z$ appears solely in the transfer function $\mathcal{H}(f_x, f_y, z)$, which exhibits unique circular symmetry in the frequency domain, as shown in Fig. 1d. To preserve this symmetry in the inverse learning process, we constrain the distance-dependent Fourier filter to a radial profile, as the diffraction behavior can be fully characterized by the radial dependency on $z$. This design reduces the number of parameters needed to represent the distance-encoded Fourier multiplier from $O(N^2)$ to $O(N)$, leading to a $10^5 \times$ reduction in the size of the distance encoding network (see Supplementary Note 2).

Figure 2a illustrates the overall architecture of SFO-solver, which consists of six hierarchical SFO blocks. Each block jointly processes spatial and Fourier features, with downsampling and upsampling operations placed between blocks to enhance multi-scale feature

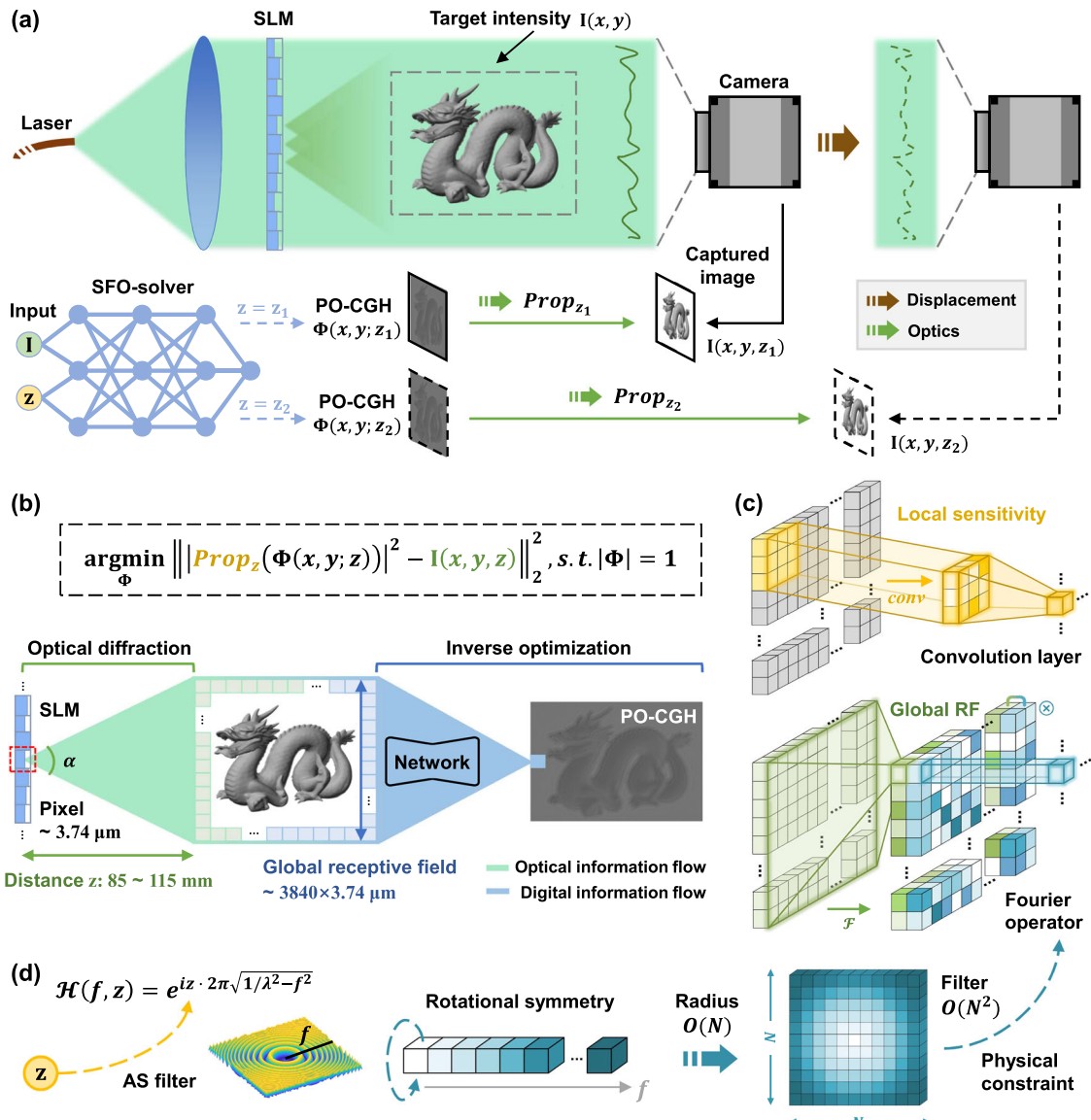

**Fig. 1 | Overview of propagation-adaptive 4K computer-generated holography using SFO-solver. a** SFO-solver takes both target intensity and propagation distance as inputs to generate the corresponding PO-CGH. Reconstructed images at various depths are captured to demonstrate the depth control capability of SFO-solver. **b** Schematic of the information flow. The digital information flow in the neural solver must align with the optical information flow of diffraction, requiring a global receptive field for typical 4K display setups. **c** Conceptual illustration of the local sensitivity provided by the convolution layer and the global receptive field achieved through the Fourier operator. **d** Rotational symmetry of the AS filter. This circular geometry is leveraged as a physical constraint in the learning framework of SFO-solver. **a, b** Images reproduced from the Stanford 3D Scanning Repository[58] (https://graphics.stanford.edu/data/3Dscanrep/).

learning, as detailed in Fig. 2c. The input distance is incorporated through a multi-layer perceptron (MLP) following a neural radiance field structure[40,41]. To enforce the circular constraint and facilitate the learning of high-frequency variations[42,43], we apply special operations such as circular mapping and Fourier embedding (see Methods 4.1), as depicted in Fig. 2b. The choice of parameters in distance encoding is based on our sampling analysis in Supplementary Note 3 and further supported by the high-frequency structures observed in the learned circular multipliers shown in Fig. 2d. A self-supervised training pipeline is adopted. Given a target intensity and a randomly-sampled propagation distance between 85 mm and 115 mm, SFO-solver predicts the corresponding PO-CGH, which is then digitally propagated to form the reconstruction pattern. Further training details are provided in Methods 4.2 and Supplementary Note 4.

## Numerical simulation and performance analysis

Figure 3 presents the simulation results for SFO-solver evaluated on 330 testing data pairs, covering diverse image styles and a continuous range of propagation distances. We compare the performance of SFO-solver with existing learning-based CGH algorithms, including Holo-encoder[28] and HoloNet[23]. As shown in Fig. 3a, SFO-solver demonstrates three key advantages. First, it supports CGH synthesis at full 4K resolution with a pixel pitch of 3.74 μm, allowing the reconstruction of images with spatially finer details. Second, SFO-solver achieves unprecedented flexibility across a 30 mm continuous depth range, whereas previous CGH networks typically require the propagation distance to remain fixed. Finally, SFO-solver achieves superior image quality, reaching up to 40 dB PSNR in numerical reconstructions, thereby demonstrating high accuracy in solving the inverse optimization problem defined in Eq. (1). Figure 3b further demonstrates SFO-

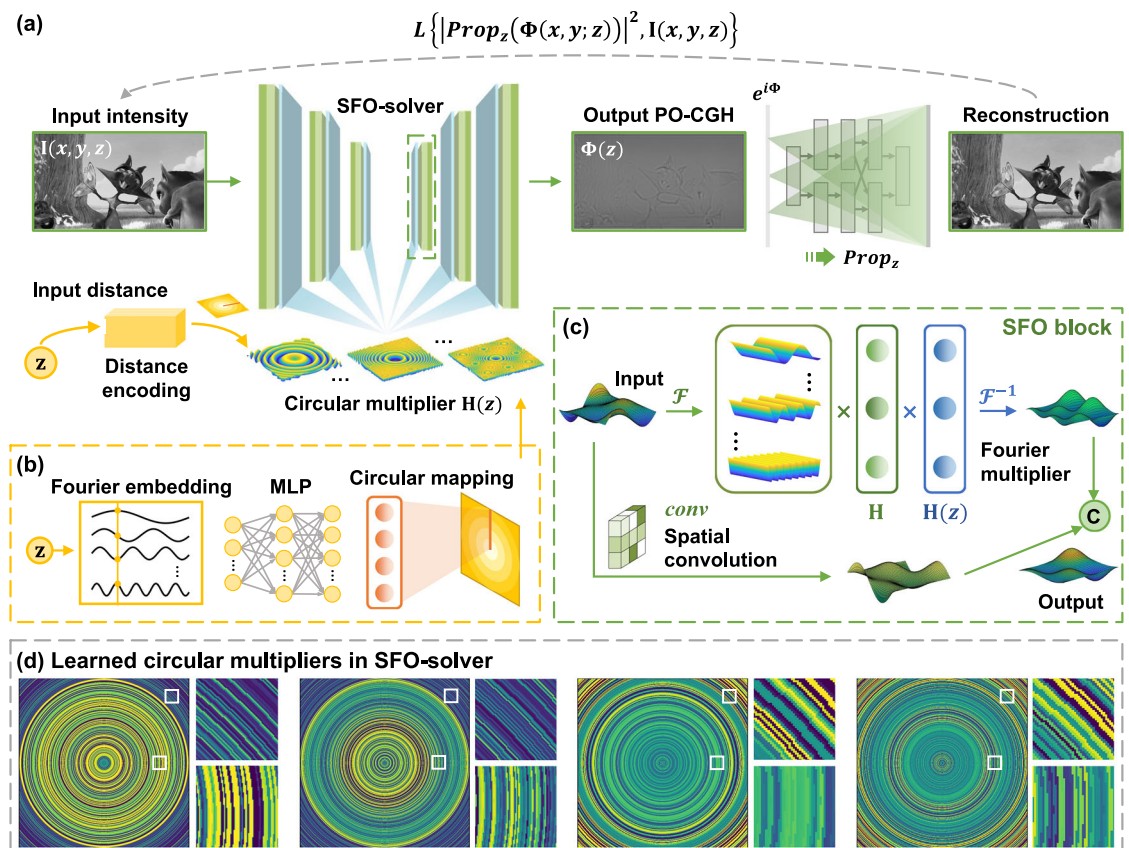

**Fig. 2 | Illustration of the SFO-solver learning framework. a** Self-supervised training pipeline of SFO-solver. Given a target intensity and propagation distance, the network predicts the corresponding PO-CGH, which is digitally propagated to compute the reconstruction loss. **b** Distance encoding scheme of SFO-solver. The input distance $z$ is processed through Fourier embedding, an MLP model and circular mapping to generate the circular Fourier multiplier. **c** Structure of the SFO block, consisting of a Fourier branch and a spatial branch, with intermediate outputs concatenated to form the final output. **d** Example visualizations of the learned circular multipliers in intermediate layers. The presence of high-frequency components validates the use of Fourier embedding and supports the parameter selection for effective distance encoding. **a** Images reproduced from www. bigbuckbunny.org (© 2008, Blender Foundation) under a Creative Commons licence (https://creativecommons.org/licenses/by/3.0/).

solver's performance consistency in synthesizing high-quality CGHs across varying propagation distances. The reconstructed images achieve an average PSNR of 39.25 dB, with the minimum value exceeding 35 dB, and an average SSIM of 0.986, with the minimum exceeding 0.97. We additionally benchmark SFO-solver against non-learning CGH methods, including the direct encoding method DPAC[18] and iterative methods such as GS[19] and SGD[23] (see Supplementary Note 5). SFO-solver achieves the best overall reconstruction quality while maintaining an average computation time of 0.157 seconds per frame. In contrast, while SGD achieves comparable PSNR, it requires approximately 100 seconds of iterative optimization to generate a single frame.

Figure 3c presents the numerical reconstructions of holograms synthesized by SFO-solver at different reconstruction distances across a 30 mm depth range. The target intensity is taken from a scene in the Big Buck Bunny animation, and the input propagation distances are selected as 85 mm, 95 mm, 105 mm, and 115 mm to better visualize the in-focus details and defocus behavior. For each hologram computed based on these inputs, the reconstructions exhibit clear image details at the corresponding focal planes, as highlighted by the close-ups in the red bounding boxes. Different levels of image defocus blur are observed at out-of-focus planes, consistent with physical diffraction. Furthermore, we notice that when a continuously shifting series of input distances is provided for a fixed target intensity, SFO-solver produces synthesized holograms

with small ripple-like variations between frames (see Supplementary Movies S1 and S2). These subtle variations of holograms precisely adjust the holograms' focal plane, further demonstrating SFO-solver's effective learning of the optical diffraction process. We further evaluate the generalization capability of SFO-solver beyond its training range by testing on propagation distances outside 85–115 mm. As detailed in Supplementary Note 6, although the training data spans only a 30 mm depth range, SFO-solver achieves stable reconstructions across a 40 mm extended depth range of 80–120 mm with minimal quality degradation. In contrast, the model trained without Fourier embedding exhibits abrupt performance collapse at the boundaries of the training range. These results confirm that SFO-solver generalizes effectively and has successfully captured the underlying physics of optical diffraction.

**Holographic projection with high spatial resolution and flexible depth control**

The spatial resolution and depth control ability of SFO-solver is experimentally demonstrated in Fig. 4, where a resolution chart is holographically projected at multiple propagation depths. Here we present only the red channel results; reconstructions for the green and blue channels are provided in Supplementary Note 7. Figure 4a shows the holograms generated by SFO-solver with different input distances: $z = 85$ mm, 95 mm, 105 mm and 115 mm, denoted as $\Phi_1$ to $\Phi_4$, respectively. The corresponding reconstructions, captured at each

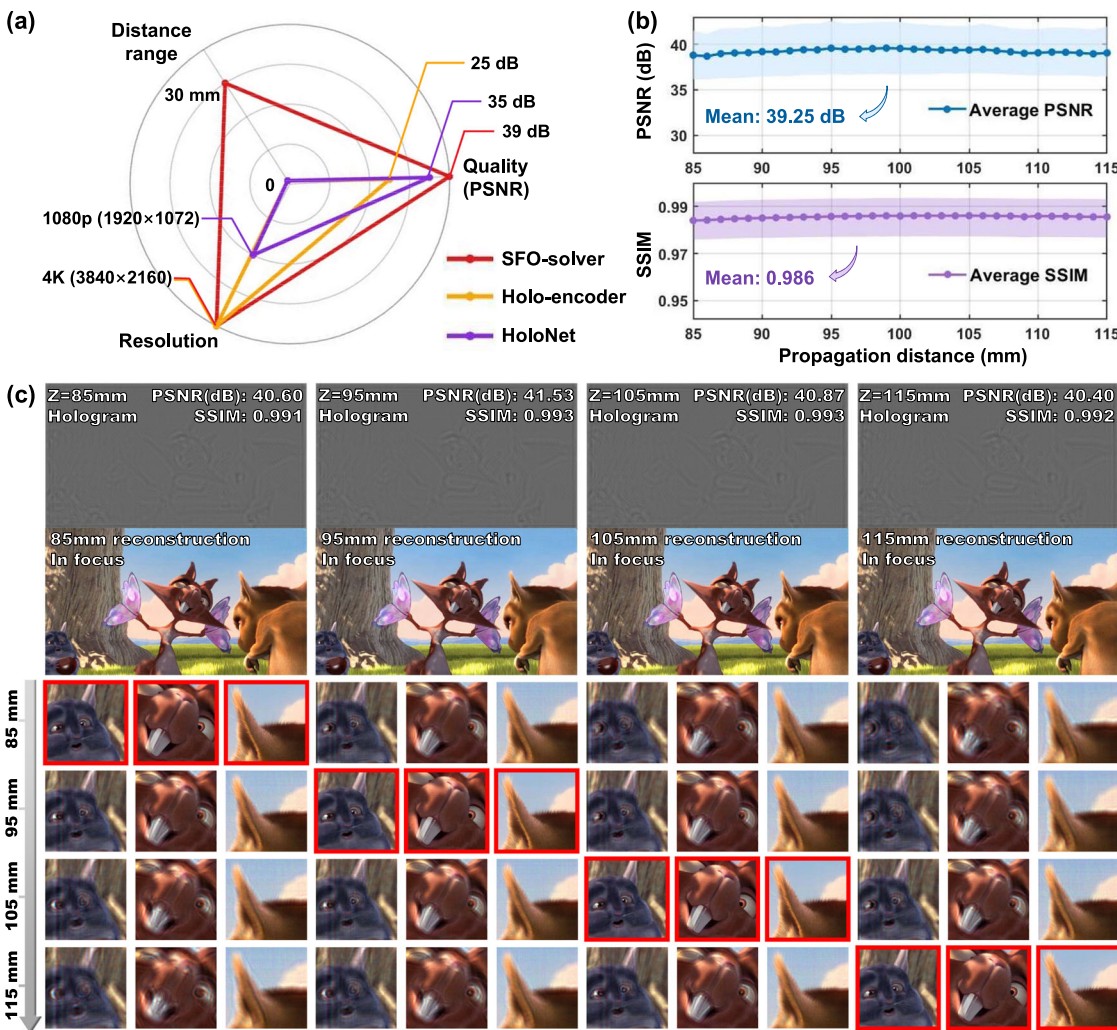

**Fig. 3 | Simulation results and performance comparison. a** Comparison between SFO-solver and existing learning-based CGH algorithms. SFO-solver demonstrates high image quality at 4K resolution with unprecedented distance range control. **b** Performance consistency of SFO-solver in synthesizing high-quality CGHs varying propagation distances. **c** Example synthesized holograms (red channel) and numerical reconstructions at 85 mm, 95 mm, 105 mm and 115 mm. Images reproduced from www.bigbuckbunny.org (© 2008, Blender Foundation) under a Creative Commons licence (https://creativecommons.org/licenses/by/3.0/).

focal plane, are displayed in columns (a1-a4) with orange and cyan bounding boxes indicating the close-up regions. Figure 4b shows the intensity line traces from the in-focus reconstructions marked with arrows, indicating that the prototype achieves a spatial resolution of 26 μm at peak-to-valley line width. Reconstructions at out-of-focus planes exhibit varying degrees of defocus blur, consistent with physical expectations. This precise depth control further enables dynamic refocusing, allowing compensation for defocus caused by axial displacement of the camera. Supplementary Movies S3 and S4 showcase this process, where holograms are computed in response to camera movement. As the camera shifts away from the original focal plane, the reconstruction becomes progressively blurred, which can be quantitatively assessed using standard image sharpness metrics[44]. By feeding the updated distance to SFO-solver, a new PO-CGH can be synthesized and uploaded to the SLM, enabling real-time refocusing of the projected pattern.

Figure 5 and Fig. 6a present the full-color holographic display of a Big Buck Bunny animation frame and the Tsinghua University emblem, enabled by SFO-solver. Along the optical axis, the observer can view these projected scenes at various focal planes. Fine image details, highlighted in red boxes, along with corresponding defocus blur, are clearly visible at different reconstruction distances. Notably,

for scenes lacking highly structured content, such as the animation scene, the out-of-focus blur exhibits relatively uniform diffusion, even though no random phases or defocus-aware constraints are applied during network training[45,46]. This natural-looking defocus resembles real optical accommodation, demonstrating the perceptual consistency of our holographic display prototype. This behavior is further attributed to the use of a high-resolution SLM and the spatial filtering of our optical system: the 3.74 μm pixel pitch enables higher spatial frequency components in the modulated wavefront, which diverge more rapidly during free-space propagation.

### Demonstration of multi-plane 3D holographic display

Leveraging SFO-solver's depth-adaptive capability, we demonstrate a multi-plane 3D holographic display system, as shown in Fig. 6b. Based on the layer-oriented method and proper phase extraction[47], PO-CGHs for multiple depth layers can be synthesized. The dual-plane CGH generation process is schematically illustrated in Fig. 6b1−6b3. First, two target intensity images and their corresponding propagation distances are spatially aligned. These paired inputs are then processed by SFO-solver, with each intensity–distance pair forming an independent batch element. The resulting PO-CGHs for each plane are subsequently

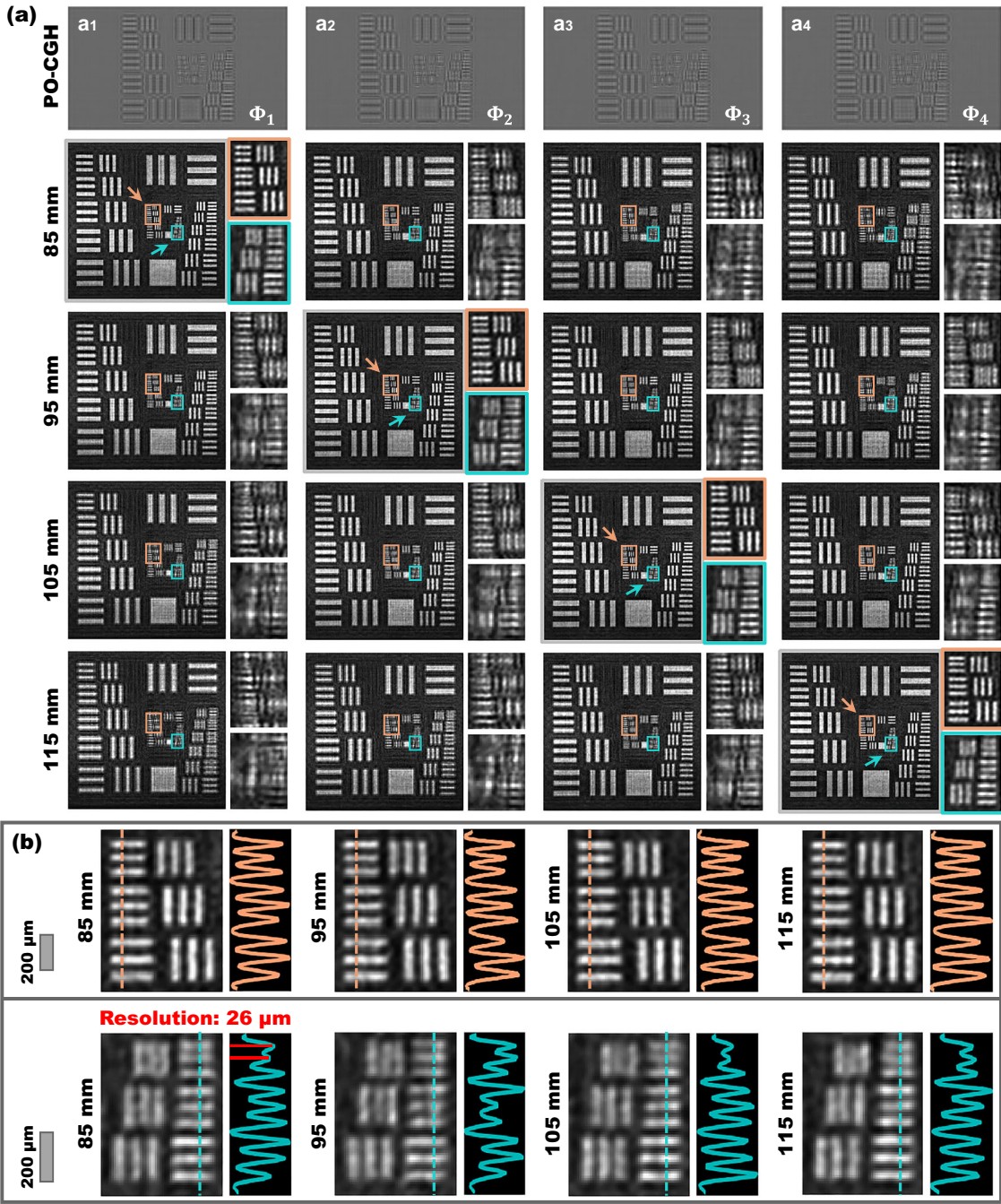

**Fig. 4 | Experimental holographic projection of a resolution chart. a** Holograms generated by SFO-solver with different input distances and corresponding optical reconstructions captured at different planes. Input distances: $\Phi_1(z = 85\,\text{mm})$, $\Phi_2(z = 95\,\text{mm})$, $\Phi_3(z = 105\,\text{mm})$, $\Phi_4(z = 115\,\text{mm})$. **b** Close-up views of the in-focus reconstructions and their intensity line traces. Scale bars indicate the actual size of the projected patterns.

merged using a phase extraction method to obtain the final PO-CGH (see Supplementary Note 8). Notably, by squeezing multiple input pairs into a single batch during SFO-solver inference, the generation of such multi-plane 3D CGHs incurs minimal additional computation time.

Figure 6b4–b6 demonstrate the experimental 3D holographic display of the 'Tsinghua' Chinese character and the 'HOLOLAB' emblem using PO-CGHs generated by SFO-solver. The holograms here only show the blue channel, while the reconstructed images are full-color results. During the experiment, the camera is fixed at 85 mm while the SLM is refreshed with holograms corresponding to specific depth configurations. By adjusting the input distances, SFO-solver

enables flexible control over the relative spatial placement of the two objects. Figure 6b4 shows both objects reconstructed at the same depth, resulting in a uniformly focused image. In Fig. 6b5, 'HOLOLAB' appears in focus at the front plane while 'Tsinghua' is defocused in the background; this configuration is reversed in Fig. 6b6. The dynamic manipulation of relative depth is further illustrated in Supplementary Movie S5, demonstrating SFO-solver's real-time depth control capability. To further validate the practical utility of SFO-solver in more complex 3D scenes, we also demonstrate holographic display of natural and overlapping objects—such as animals, vehicles, insects, and poker cards arranged across distinct depth planes—as detailed in Supplementary Note 9. These results confirm not only the robustness

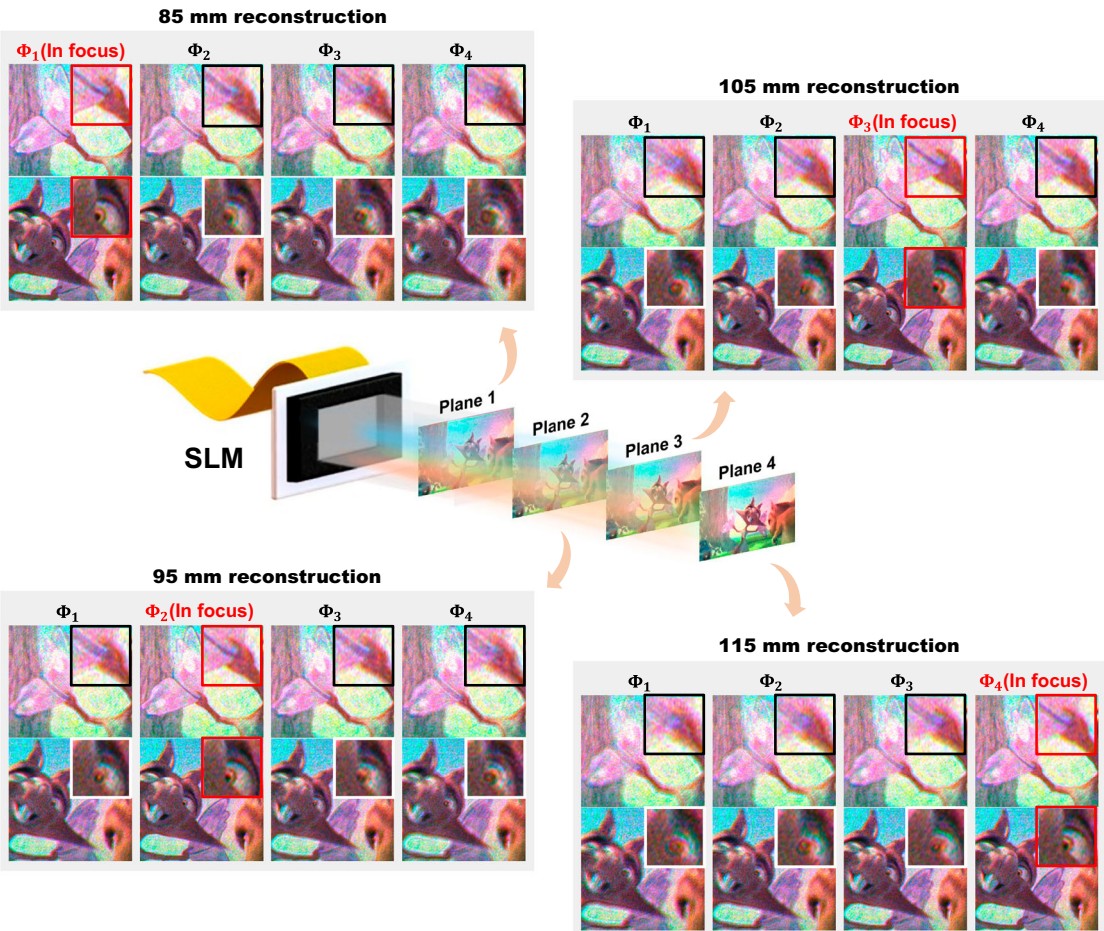

**Fig. 5 | Experimental demonstration of full-color holographic display at different focus planes.** Holograms are generated using SFO-solver with different input distances and the close-up photographs are captured at different depths.

Images reproduced from www.bigbuckbunny.org (© 2008, Blender Foundation) under a Creative Commons licence (https://creativecommons.org/licenses/by/3.0/).

of SFO-solver in handling occlusion and layered depth, but also its strong potential for enabling realistic, interactive volumetric holographic displays.

## Discussion

SFO-solver introduces a physics-constrained deep learning framework for propagation-adaptive computer-generated holography, where the core physical principles of optical diffraction are systematically embedded into both the network architecture and the learning pipeline. By aligning the model's receptive field with the optical information flow and encoding the propagation distance via a circularly symmetric Fourier representation, SFO-solver effectively bridges the flexibility of traditional iterative solvers with the efficiency of neural inference. Comparative evaluations against both learning-based and non-learning CGH algorithms confirm its unique advantage in simultaneously achieving high spatial fidelity, fast computation, and flexible depth control. While the recently proposed "Conditional Neural Holography"[48] shares the same goal of depth-adaptive CGH, SFO-solver offers a more principled alignment with the underlying physics of optical diffraction, yielding consistently higher-quality reconstructions in both simulations and experiments. Our demonstrations of dynamic holographic projection and focus-adjustable 3D displays further underscore the practical utility of SFO-solver in various user-specified holographic systems.

The choice of an 85–115 mm working distance in our experiments is informed by both practical and theoretical considerations. First, the 100 mm central distance is consistent with typical configurations reported in prior literature, where the spatial footprint of optical elements commonly dictates working distances in the range of 100–200 mm. Second, the selected 30 mm depth span used here is sufficient to produce perceptually meaningful defocus cues under a 4K SLM with 3.74 μm pixel pitch, as verified in our optical experiments. As shown in Eq. (S4) of Supplementary Note 1, the required sampling density increases quadratically as the SLM pixel size decreases due to the $O(f^2)$ frequency variation in the AS filter. This scaling implies that, under the same sampling condition used for distance encoding, SFO-solver could maintain comparable focus control precision across a depth range exceeding 120 mm when deployed on a 1080p commercial SLM with a typical 8 μm pixel pitch. Further extension of the depth range is also possible by increasing the sampling density, albeit at the cost of increased GPU memory and computational overhead. Alternatively, the use of relay optics could optically expand the usable depth range, although the x-y spatial scaling would need to be adjusted accordingly along with the depth position.

Beyond spatial depth, SFO-solver's physics-constrained design can be extended to encode other physical variables such as wavelength. Since the AS filter exhibits circular symmetry in both distance and wavelength, a unified model could be built to support wavelength-adaptive holography. However, this flexibility is less critical in practice, as most full-color CGH systems use discrete RGB laser sources, making separate training of three SFO-solvers a simpler and more reliable solution. For this reason, wavelength conditioning was not pursued in the current study.

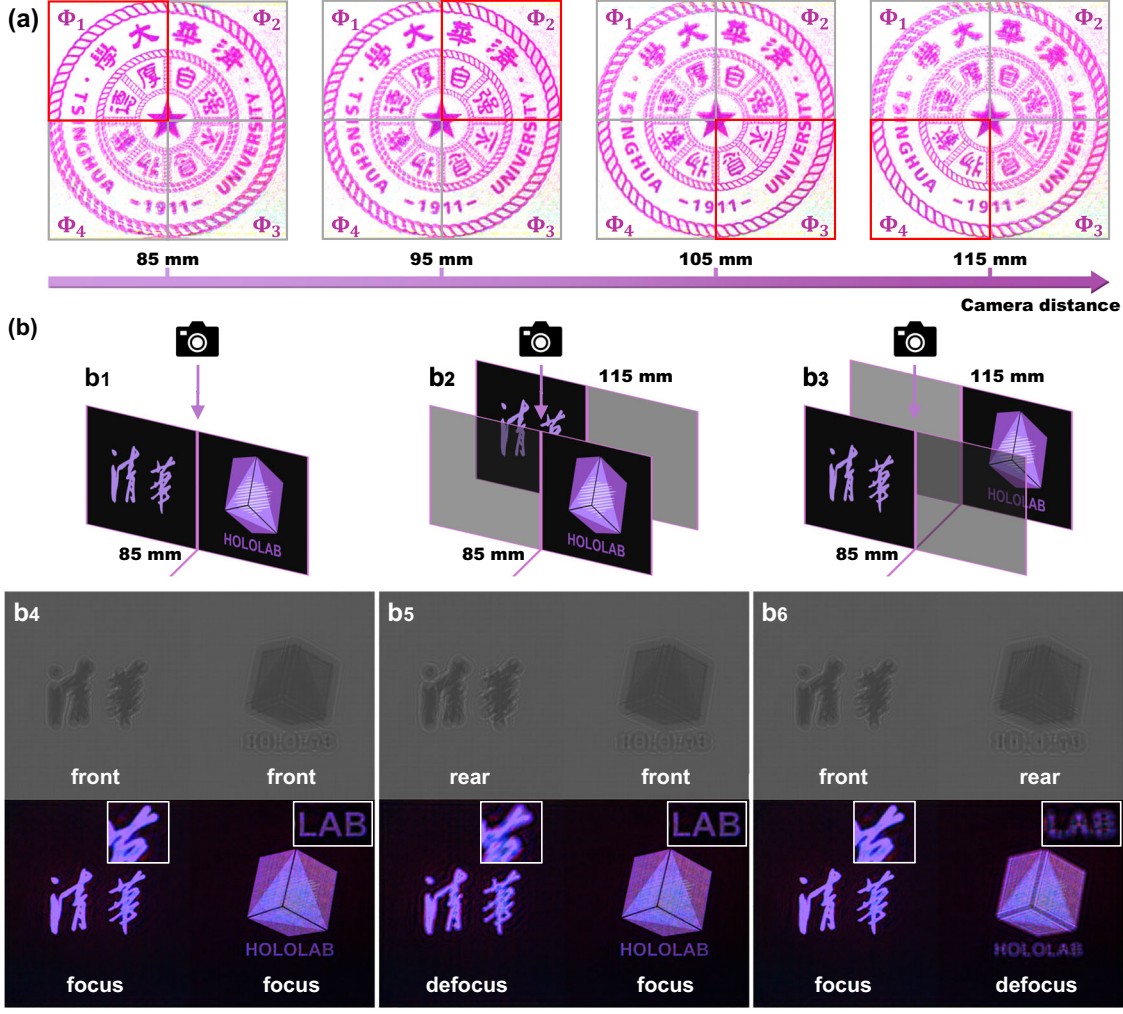

**Fig. 6 | Experimental demonstration of flexible-focus 2D projection and multi-plane 3D holographic display using SFO-solver. a** Demonstration of depth-adaptive control for single-plane 2D projections. Reconstructed images at varying depths confirm that focus and defocus effects are accurately rendered by PO-CGHs

generated using different input distances. **b** Demonstration of focus control in 3D display. The relative depth position of objects can be adjusted according to different specifications.

SFO-solver's strong generalization, high-resolution synthesis, and flexible depth control make it a promising solution for user-oriented dynamic holographic displays. Further improvements in experimental performance may be achieved through optical model corrections and speckle suppression techniques, such as camera-in-the-loop learning[49,50] and multiplexing[51–53]. While our current implementation of multi-plane display adopts a layer-based strategy, future extensions incorporating focal stack representations could enable more immersive volumetric effects. These directions, alongside continued efforts to optimize inference speed and network efficiency, highlight the scalability and versatility of the SFO-solver framework for a broad spectrum of advanced holographic applications in next-generation human–computer interaction.

## Methods
### Fourier embedding for SFO-solver's distance encoding
To encode the input propagation distance in the Fourier branch of SFO-solver, we construct a neural radiance field that maps the scalar distance to a circular filter. However, standard MLPs, as commonly used in neural radiance fields, are known to struggle with learning high frequency features[42,43]. To address this limitation, we introduce a

Fourier embedding operation that guides the network in learning the high-frequency structure of the Fourier filter. Given an input distance $z$, the mapping function can be written as:

$$\zeta(z) = \left(\sin(\omega_{min}z), \cos(\omega_{min}z), \ldots, \sin(\omega_{max}z), \cos(\omega_{max}z)\right) \quad (2)$$

where $\zeta: \mathbb{R} \to \mathbb{R}^{2N}$ is a fixed Fourier mapping that projects the scalar input into a higher-dimensional space. The embedded tensor $\zeta(z)$ is composed of sinusoidal components with frequencies ranging from $\omega_{min}$ to $\omega_{max}$. Since the distance encoding effectively models the functional relationship between propagation distance and the spatial frequency response of the AS filter, the frequency band $[\omega_{min}, \omega_{max}]$ is selected to match the estimated frequency variation of the AS propagator. This alignment is illustrated in Supplementary Fig. S3a. The frequency interval $\Delta\omega = (\omega_{max} - \omega_{min})/N$ is determined as twice the maximum local frequency of the AS propagator, following the Nyquist sampling criterion. In our implementation, the number of embedding frequencies $N$ is set to 689, 560 and 484 for the red, green and blue channels, respectively. The values of $\omega_{min}$ and $\omega_{max}$ equal to the minimum and maximum frequency with respect to distance $z$ in the AS filter $H(z)$ (see Supplementary Note 3 for a full derivation of

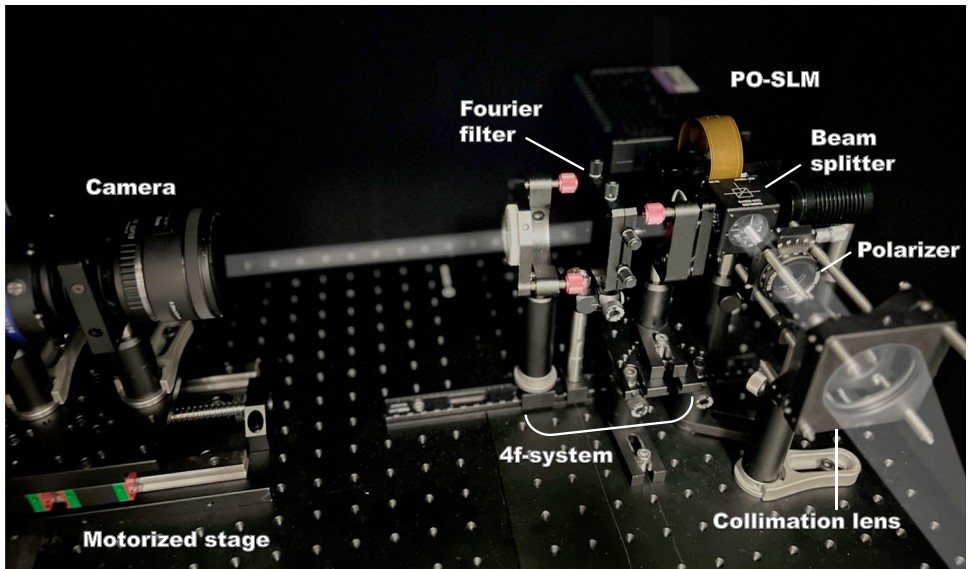

**Fig. 7 | Photograph of the experimental setup.** Collimated RGB laser beams illuminate the phase-only spatial light modulator (PO-SLM) through a linear polarizer and a beam splitter. The modulated wavefront is then directed into a 4f system for diffraction order selection. Reconstructed images are captured by a camera mounted on a motorized translation stage for depth-resolved imaging.

parameter selection).

$$\omega_{\min} = 2\pi \sqrt{\frac{1}{\lambda^2} - 2\left(\frac{1}{2\Delta x}\right)^2}; \; \omega_{\max} = \frac{2\pi}{\lambda} \quad (3)$$

where $\lambda$ is the optical wavelength and $\Delta x$ is the SLM's pixel pitch size. We conduct the ablation studies regarding the Fourier embedding operation in Supplementary Note 6, which confirm that it allows SFO-solver to generalize to a broader range of unseen propagation distances. This validates the embedding's role in enabling SFO-solver to effectively learn the underlying physics of optical diffraction.

### Data preparation and training configuration

In the training process the DIV2K dataset[54] is properly resized to the targeted resolution for network training. For each input intensity, ten distances are randomly sampled between 85 mm and 115 mm to form the input data pairs. In total there are 8000 intensity-distance data pairs for training and 330 data pairs with different image styles for testing (see Supplementary Note 4 for detailed illustrations of the training and testing data). The propagation operator that we implement in the self-supervised learning framework follows the band-limited AS method[31].

$$Prop_z(\Phi) = \iint \mathcal{F}(\Phi) \mathcal{H}\left(f_x, f_y, z\right) e^{i2\pi(f_x x + f_y y)} df_x df_y,$$

$$\mathcal{H}\left(f_x, f_y, z\right) = \begin{cases} e^{\frac{i2\pi z}{\lambda}\sqrt{1-(\lambda f_x)^2-(\lambda f_y)^2}}, & \text{if } f_x^2 + f_y^2 < \min\left\{\frac{1}{\lambda^2}, f_b^2\right\}, \\ 0 & \text{otherwise.} \end{cases} \quad (4)$$

Here $\mathcal{F}(\cdot)$ denotes the Fourier transform, $f_x, f_y$ are spatial frequencies, and $\lambda$ is the wavelength. $f_b$ denotes the band constraint applied in band-limited AS method (see Supplementary Note 10 for more discussions on the band constraint). In the self-supervised framework shown in Fig. 2a, we employ a loss function that measures both data fidelity and perceptual quality between the target intensity $I$

and reconstruction $\hat{I}$ to train SFO-solver.

$$\begin{aligned} L(\hat{I}, I) = 1 \;\; &- \frac{\sum_k (\hat{i}_k - \bar{\hat{i}})(i_k - \bar{i})}{\sqrt{\sum_k (\hat{i}_k - \bar{\hat{i}})^2 \sum_k (i_k - \bar{i})^2}} && \dots \text{NPCC loss} \\ &+ \beta_1 ||\hat{I} - I||_2^2 && \dots \text{MSE loss} \\ &+ \beta_2 ||\nabla\hat{I} - \nabla I||_1 && \dots \text{Gradient loss} \end{aligned} \quad (5)$$

Here, the negative Pearson correlation coefficient (NPCC) and the mean square error (MSE) are used to ensure data fidelity, while the total variation difference helps enhance the perceptual quality by minimizing gradient-domain loss[55]. The corresponding weight parameters, $\beta_1, \beta_2$ are adjusted accordingly. Experiments have shown that setting $\beta_1$ and $\beta_2$ to small values at the first stage of training ($\beta_1 = \beta_2 = 0.1$, learning rate set as $10^{-3}$) helps the network learn more general features and speeds up convergence. In the second stage of network finetuning, we set $\beta_1 = \beta_2 = 1$ and the learning rate to be $10^{-4}$. Both network parameters and digital propagation in the training process are implemented on an NVIDIA V100 GPU. The whole training process took approximately 40 hours.

### Experimental details

The holographic display prototype for optical experiments is shown in Fig. 7. A combined RGB laser diode (red: 638 nm, green: 520 nm, blue: 450 nm) is first collimated and linearly polarized before illuminating on the SLM (HOLOEYE GAEA−2). Blazed grating phase is superimposed onto the uploaded holograms to spatially separate the non-modulated reflection light and the modulated diffraction light of the SLM at the Fourier plane. The diffracted light is then filtered through a 4 f system composed of two achromatic lenses and an adjustable slit positioned at the Fourier plane. This spatial filter is manually aligned to match the bandwidth constraint used during training of the SFO-solver[56], ensuring consistent frequency support across simulation and experiment. To capture reconstructions at varying depths, the camera is mounted on a motorized translation stage that moves along the optical axis. This setup allows high-precision imaging at specific propagation distances for quantitative evaluation of the depth control and image quality.

## Data availability
All processed test data in this study have been deposited in the Google Cloud Drive under accession https://drive.google.com/drive/folders/17h8pox1Wh5M2rPspZ6HLve3HG38B9BC0?usp=drive_link and can be accessed following instructions at Ref. 57. The raw data can be obtained from the authors upon request.

## Code availability
The original code has been released at Ref. 57 and is publicly available.

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

## Acknowledgements
This work was supported by National Natural Science Foundation of China 62235009, 62035003 and 62441613 (LC). NL thanks the group of Spark Project at Tsinghua University for helpful discussions.

## Author contributions
N.L. and L.C. conceived the idea; N.L. developed the algorithm and trained the neural network; N.L. conducted the experiments with the assistance of K.L. and Y.Y.; Y.P. and L.C. provided mentoring support and valuable instructions; N.L. wrote the manuscript with input from all authors.

## Competing interests
The authors declare no competing interests.
