## [Transparent Peer Review file · Nature Communications]

Propagation-adaptive 4K computer-generated holography using physics-constrained spatial and Fourier neural operator

Corresponding Author: Professor Liangcai Cao

Version 0:

Reviewer comments:

Reviewer #1

(Remarks to the Author)

The manuscript presents a novel deep learning-based framework, SFO-solver, for propagation-adaptive 4K computer-generated holography (CGH). The authors demonstrate a significant advancement in CGH synthesis by incorporating both target intensity and propagation distance as inputs, enabling flexible depth control and high-speed hologram generation. The proposed method leverages physics-constrained encoding using Spatial and Fourier neural Operators, achieving impressive results in terms of image quality as well as computational efficiency. The experimental validation, including 2D holographic projections and multi-plane 3D displays, showcases the potential of SFO-solver for dynamic refocusing and user-interactive holographic applications. While the work is innovative and well-executed, there are several critical issues that need to be addressed to strengthen the manuscript and ensure its suitability for publication in Nature Communications.

1. The manuscript compares SFO-solver with non-learning CGH algorithms (e.g., DPAC, GS, SGD) and some learning-based methods (e.g., Holo-encoder, HoloNet). However, it does not provide a comprehensive comparison with the most recent state-of-the-art learning-based CGH methods, particularly those that also aim to address the propagation distance flexibility, such as "Conditional neural holography: a distance-adaptive CGH generator" (<https://arxiv.org/abs/2411.04613>). A more thorough comparison with these methods, including their computational efficiency, image quality, and depth control capabilities, would help to better position SFO-solver within the current landscape of CGH research. In addition, when comparing with Holo-encoder and HoloNet, it is also suggested to add the results of hologram reconstructions while Fig. 3 only shows the reconstructions from the proposed hologram.
2. The manuscript introduces Fourier embedding and circular symmetry as key components of SFO-solver's physics-constrained encoding. While the authors provide some theoretical justification, the practical impact of these components on the overall performance of SFO-solver is not fully explored. Specifically, how does the circular symmetry constraint improve the accuracy or efficiency of the network? A more detailed ablation study or sensitivity analysis would help clarify the contribution of these components to the overall performance of SFO-solver.
3. The manuscript demonstrates the multi-plane 3D holographic display using simple binary patterns (e.g., "Tsinghua" characters and "HOLOLAB" emblems). While these results effectively illustrate the depth control capability of SFO-solver, they do not fully showcase the method's ability to handle complex, natural scenes with fine details and textures. To strengthen the manuscript and provide a more comprehensive evaluation, the authors should include additional experiments using natural images (e.g., the resolution chart or animation scenes used in the 2D projection experiments) for multi-plane 3D display. This would demonstrate the robustness of SFO-solver in handling real-world scenarios with intricate details and varying depth structures. Additionally, it would help readers better understand the practical applicability of the method in more complex holographic displays.
4. The manuscript demonstrates the performance of SFO-solver within a propagation distance range of 85 mm to 115 mm. However, the rationale for selecting this specific range, particularly the choice of 85 mm as the minimum distance, is not clearly explained. Given that the authors mention in Fig. 1 that the diffraction PSF covers an area as large as the whole image, it is crucial to clarify how this range was determined and whether it is related to the optical setup or the limitations of the method.
5. The manuscript lacks crucial information regarding the dataset used for training the SFO. Regarding the dataset, the authors only mention using the DIV2K dataset, it would be highly beneficial for the readers if the authors could showcase

samples from the training set. Visualizing the input intensity images and their corresponding distances would help the reviewers. How about the test performance to other nature image dataset? Please provide more test details for dataset diversity performance.

The manuscript presents advancement in CGH synthesis with its propagation-adaptive framework and physics-constrained encoding. However, to meet the high standards of Nature Communications, the authors need to address the aforementioned issues. With these revisions, the manuscript has the potential to make a substantial contribution to the field of computer-generated holography.

(Remarks on code availability)

Reviewer #2

(Remarks to the Author)

This study developed a neural network-based SFO-Solver and showed that it can generate high quality 4K CGH at high speed. It is also noteworthy that it was shown to be able to generate CGHs at arbitrary locations within a depth of 30 mm. This is because conventional neural network-based CGH calculations need to be re-trained when computational conditions, such as depth, change. The results of the simulation and optical experiments are fascinating. The manuscript is well written. However, I have the following concerns.

The authors claim that one advantage of SFO-Solver is that it can perform fast CGH calculations. However, I am quite sceptical about this claim. The authors showed that a 4K CGH can be computed in 0.16s. The GPU used was v100. I computed a 4K CGH on a GPU (RTX4060Ti) as a trial, and the computation time was almost the same as 0.16s. The GPU I used performed worse than the v100. Furthermore, it should be noted that my calculations were done with zero padding, so the actual calculations were done at 8K. At propagation distances where zero padding is not required, SFO-Solver may be considerably slower than a typical calculation. Of course, I understand that SFO-solver can achieve high image quality. However, the DPAC shown in Fig S3, for example, shows a reasonably high image quality of 25 dB, and for large propagation distances, the difference in image quality between SFO-solver and DPAC would be even smaller.

Furthermore, in what computing environment were Fig S3's GS, DPAC and SDG run - SDG was probably run on a GPU, but what about GS and DPAC? In common with the concerns mentioned earlier, the computation time for DPAC seems to be slower.

What type of holographic display is SFO-solver targeting - is it a Near-eye display?

The current SFO-solver can produce CGH with a propagation distance of 85 mm to 115 mm. Can this range be extended further? If it cannot be widened, the reasons for this should be discussed.

The current SFO-solver requires computation time proportional to the number of layers to perform layer hologram-like calculations. This seems to sacrifice computation time.

The SFO-Solver can freely shift the propagation distance. On the other hand, the authors show colour reproduction results, but can the SFO-Solver also freely set the wavelength? Or did the SFO-Solver with colour reproduction train three neural nets corresponding to RGB?

" f_b^2 " in Equation 4 is undefined.

I feel that one of the factors that makes this paper difficult to read is the use of abbreviations. For example, AO, PO, CGH, AS, L2O, DPAC, PSF, RF, MLP, NPCC, MSE, TV. And DPAC and RF are used only twice in the paper. The use of abbreviations should be kept to a minimum. L2O is also defined twice. The paper should be written with care.

Periods attached to the word "Figure" should be removed, as in "Figure. 4(a)". It is necessary to write the paper with care.

(Remarks on code availability)

Version 1:

Reviewer comments:

Reviewer #1

(Remarks to the Author)

The author made substantial revisions based on my comments, which significantly enhanced the quality of the article. I think it can be considered for publication now.

(Remarks on code availability)

The code provided by authors in github is highly valuable and serves as a great reference for researchers in the community of computer holography.

Reviewer #2

(Remarks to the Author)

The authors have thoroughly addressed all reviewer comments and have made corresponding revisions to the manuscript to improve its clarity, technical depth, and presentation. One of the key concerns raised during the review process was the lack of explanation regarding the novelty and significance of the proposed SFO-solver. In the revised manuscript, the authors have clearly highlighted the unique aspects of their approach, particularly the use of both spatial and Fourier neural operators in a physically informed manner. They elaborated on how their method leverages global information flow and circular symmetry—two fundamental characteristics of optical diffraction—which contributes to the solver's adaptability and enhanced performance.

Another important point raised by the reviewers concerned the flexibility of the method with respect to varying propagation distances. The authors have convincingly demonstrated that their model supports propagation-adaptive CGH synthesis, as opposed to prior models that were constrained to fixed distances. To support this, they included new experimental results that show consistent image quality across a 30 mm depth range, thereby substantiating the claims of generalizability and robustness. Moreover, they provided additional details on the network input structure and how propagation distance is integrated into the inference process, improving the transparency and interpretability of their model.

The runtime performance was another area of interest. The authors presented comprehensive performance benchmarks, including comparisons with existing methods. Their results show that the SFO-solver achieves high-speed 4K CGH synthesis at 0.16 seconds per frame, which is a substantial improvement over previous approaches. These updates make the practical relevance of the method more evident, especially for real-time and interactive applications.

In terms of presentation, the revised manuscript shows clear improvement. The authors have rewritten several sections for better clarity, included more informative figures, and improved the flow of the narrative. The abstract and conclusion now better reflect the contributions and potential impact of the work. The supplementary material has also been updated to provide deeper insights into the implementation details, hyperparameters, and experimental setup, making the study more reproducible.

Overall, the authors have made a strong effort to address all concerns raised during the review. The revised manuscript is now well-organized, technically sound, and provides a more compelling case for the utility and novelty of the proposed method. The paper is now in a publishable state and is expected to make a valuable contribution to the field of computer-generated holography and computational optics.

(Remarks on code availability)

Revision feedback for reviewers

Reply to Editors & Reviewers

We would like to extend our sincere gratitude to all reviewers and editors for their time and efforts in reviewing and processing our manuscript. We are particularly grateful for the insightful comments and valuable suggestions that have significantly contributed to enhancing the manuscript. In response to this feedback, we have made substantial revisions to the manuscript, and all changes are marked in the revised manuscript. Please find below point-by-point responses to the editors and reviewers, where we present their comments in **blue** and highlight our revisions in **red** in the revised manuscript.

Best regards,

All authors

Reviewer #1 Point 1.1: The manuscript compares SFO-solver with non-learning CGH algorithms (e.g., DPAC, GS, SGD) and some learning-based methods (e.g., Holo-encoder, HoloNet). However, it does not provide a comprehensive comparison with the most recent state-of-the-art learning-based CGH methods, particularly those that also aim to address the propagation distance flexibility, such as “Conditional neural holography: a distance-adaptive CGH generator”. (<https://arxiv.org/abs/2411.04613>) A more thorough comparison with these methods, including their computational efficiency, image quality, and depth control capabilities, would help to better position SFO-solver within the current landscape of CGH research.

Reply: Reviewer #1 kindly suggested a comparison with the recent “Conditional Neural Holography”¹, which was initially shared on arXiv when at the time of our manuscript submission and has recently been published in Optics Express. To our knowledge, this remains the only publicly available work to date that shares a similar objective to our SFO-solver: enabling depth-adaptive CGH using neural networks. While both approaches aim to address similar core challenges, we respectfully underscore three fundamental differences that distinguish our method in terms of physical interpretability, modeling strategy, and practical performance.

First, although “Conditional Neural Holography” achieves depth-adaptive holography, its design appears to lack strong physical grounding. The network backbone is built on traditional CNNs, which inherently have limited receptive fields (see the first paragraph of the revised manuscript Results 2.2 and Supplementary Note 1). This may limit the network’s ability to recover the global optical information required by the inverse diffraction process. Furthermore, their distance encoding relies on a single low-frequency zone plate representation. In contrast, our analysis (see revised Supplementary Note 3) shows that the distance dependence of the diffraction kernel (based on the angular spectrum solution of the Helmholtz equation) exhibits high-frequency oscillations (This high-frequency structure is also visualized in the revised manuscript Fig. 2(d) and Supplementary Note 3). The authors of “Conditional Neural Holography” reported an average PSNR of 28.64 dB in their paper. We suggest that this lower image quality could potentially stem from their misalignment of both information flow and frequency range. In contrast, SFO-solver employs a Spatial and Fourier Operator backbone with distance

encoding that explicitly captures high-frequency distance dependence, ensuring physics consistency and achieving a significantly higher average PSNR of 39.25 dB.

Second, while “Conditional Neural Holography” demonstrates distance adaptability across a 100 mm range, its encoding strategy is discrete. Specifically, their network learns to map a finite set of zone plates to corresponding CGHs (100 points for their reported 28.64 dB results), which essentially only requires the network to memorize these 100 distances. Our SFO-solver, by contrast, learns a continuous and physically informed distance-to-Fourier-filter mapping, which allows generalization across unseen distances, as verified in our depth extension experiments (see the second paragraph of the revised manuscript Results 2.3 and Supplementary Note 6). Moreover, because we’re using a 4K SLM with 3.74 μm small pixel pitch, our 30 mm distance range translates to over 120 mm on a typical 1080p SLM with 8 μm pixel pitch (see the discussions of AS filter’s frequency behavior in Supplementary Note 3).

Third, as shown in Fig. R1, “Conditional Neural Holography” exhibits a quite noticeable quality gap between its simulated reconstructions and optical experiments, suggesting limited robustness in practical applications. In comparison, our results show much better alignment between simulation and real-world holographic display. Our experimental demonstration of dynamic refocusing and depth-adjustable 3D display further proves the broad applicability of SFO-solver.

[editorial note: figure redacted]

Fig. R1. Optical display results reported in “Conditional Neural Holography”¹

Also, since the authors of “Conditional Neural Holography” didn’t write a user guidance nor released their model weights in their Github page, we feel less inclined to repeat their work as an official benchmark comparison. Instead, we listed a comparison summary table between SFO-solver and “Conditional Neural Holography” based on their statements in the published paper for reviewers’ reference. Generally, while “Conditional Neural Holography” represents an important first step toward depth-adaptive CGH, our method offers a more physically interpretable framework with better reconstruction accuracy,

stronger generalization, and greater experimental reliability. We have clarified these points in the revised Discussion section of the manuscript.

Table R1. Comparison between SFO-solver and “Conditional Neural Holography”

	SFO-solver	Conditional Neural Holography
Model backbone	Spatial and Fourier operator	Convolution layer
Distance encoding	Multiple neural-represented Fourier filters (high frequency)	Single discrete zone plate (low frequency)
Distance range / SLM resolution	30 mm / 4K (3,840×2,160) extended to 40 mm	100 mm / 2,048×1,024
Physical interpretability	Verified through extension on unseen distances	\
Reported simulation quality	Average PSNR: 39.25 dB	Average PSNR: 28.64 dB
Experiment quality	Consistent with simulation	Large quality gap between simulation and experiments
Application demonstrations	2D display, dynamic refocusing, Depth-adjustable 3D display	2D display

Finally, we would like to emphasize the positioning of SFO-solver within the CGH research community. As stated in the opening paragraph of the revised manuscript Introduction Section, CGH algorithms can generally be categorized into two groups: non-learning and learning-based methods. For traditional non-learning approaches (except for DPAC—see our *response to Reviewer #2, Point 2.2* for further discussion), there exists an inherent trade-off between image quality and computational runtime. Achieving high-quality holograms typically requires many iterative steps, resulting in substantial latency. In contrast, recent advances in deep learning have effectively addressed this limitation, enabling the synthesis of high-quality CGHs with significantly reduced inference time. This context has been established in the well-known “Neural Holography”² by Peng et al. and is now widely recognized within the CGH research community. Although deep learning offers a compelling alternative to traditional CGH algorithms, it typically suffers

from a critical limitation—lack of flexibility—which is precisely the core motivation of our work. For a comprehensive benchmark, we include representative methods from both non-learning (classic GS, SGD³: 423 citation, DPAC⁴: 847 citations) and learning-based categories (HoloNet³: 423 citations, Holoencoder⁵: 195 citations), and demonstrate that SFO-solver combines the speed and quality of learning-based approaches with the adaptability of traditional algorithms. We respectfully note that despite the existence of “Conditional Neural Holography”, our SFO-solver remains a pioneering contribution because of due to its physics-constrained modeling framework, strong model performance, and versatile demonstrations across multiple holographic applications.

Reviewer #1 Point 1.2: The manuscript introduces Fourier embedding and circular symmetry as key components of SFO-solver’s physics-constrained encoding. While the authors provide some theoretical justification, the practical impact of these components on the overall performance of SFO-solver is not fully explored. Specifically, how does the circular symmetry constraint improve the accuracy or efficiency of the network? A more detailed ablation study or sensitivity analysis would help clarify the contribution of these components to the overall performance of SFO-solver.

Reply: Reviewer #1 kindly requested to clarify the contribution of circular symmetry and Fourier embedding in the network’s performance. We appreciate this insightful question and have done some additional work to strengthen the manuscript. For the circular constraint we conduct model parameter count to show the necessity of such operation. We have revised **Results 2.2 of the revised manuscript (the second paragraph)** and added a **Supplementary Note 2** to better explain the physical motivation for the circular constraint and its role in reducing parameter counts. Specifically, we estimate that the circular constraint leads to a 10^5 -fold reduction in the model size of MLP-based distance encoder (detailed structure in **Supplementary Fig. S2**). Without this constraint, implementing the distance encoder would require an estimated 20 TB of GPU memory, whereas the circular constraint reduces this requirement to just 80 MB—well within the capacity of most commercially available GPUs.

We also conducted detailed ablation studies of Fourier embedding in Supplementary Note 6, which confirm that it not only helps improve model performance but also allows SFO-solver to generalize to a broader range of unseen propagation distances. The visualization of the learned high-frequency Fourier filter in the revised manuscript Fig. 2(d) and Supplementary Fig. S4 also validates the embedding's role in enabling SFO-solver to effectively learn the underlying physics of optical diffraction.

Reviewer #1 Point 1.3: The manuscript demonstrates the multi-plane 3D holographic display using simple binary patterns (e.g., "Tsinghua" characters and "HOLOLAB" emblems). While these results effectively illustrate the depth control capability of SFO-solver, they do not fully showcase the method's ability to handle complex, natural scenes with fine details and textures. To strengthen the manuscript and provide a more comprehensive evaluation, the authors should include additional experiments using natural images (e.g., the resolution chart or animation scenes used in the 2D projection experiments) for multi-plane 3D display. This would demonstrate the robustness of SFO-solver in handling real-world scenarios with intricate details and varying depth structures. Additionally, it would help readers better understand the practical applicability of the method in more complex holographic displays.

Reply: Reviewer #1 kindly suggested further demonstration of SFO-solver with more realistic 3D contents. We agree that demonstrating performance with more complex scenes is valuable to demonstrate the practical applicability of SFO-solver. In Supplementary Note 9, we added experiments featuring occluded and layered natural 3D scenes (Supplementary Fig. S12: natural images such as animals, insects and vehicles, Supplementary Fig. S13: A poker cards scene with 3D overlap). These results show that SFO-solver not only supports depth adjustment for simple multi-plane scenes but also extends to more challenging 3D layouts, showing SFO-solver's great versatility. We also acknowledge that our current layer-based method has some limitations in handling 3D scenes and include some discussions for future improvements in the revised manuscript Discussion section.

Reviewer #1 Point 1.4 The manuscript demonstrates the performance of SFO-solver within a propagation distance range of 85 mm to 115 mm. However, the rationale for selecting this specific range, particularly the choice of 85 mm as the minimum distance, is not clearly explained. Given that the authors mention in Fig. 1 that the diffraction PSF covers an area as large as the whole image, it is crucial to clarify how this range was determined and whether it is related to the optical setup or the limitations of the method.

Reply: Both reviewers raised questions about why the working range of 85–115 mm was selected and whether it is sufficient. We agree with the reviewers that a thorough reasoning of the experimental parameters is crucial for the reader's understanding of our manuscript, and we are grateful for their constructive suggestions. We addressed this in the second paragraph of the revised manuscript Discussion section, noting that the central distance of 100 mm is consistent with prior literature, where the spatial footprint of optical elements commonly dictates working distances in the range of 100–200 mm. The 30 mm depth range provides sufficient perceptual defocus cues when using a 4K SLM with 3.74 μm pixel pitch, as verified in our optical experiments. Also, it's important to note that the 30 mm depth range that we choose with 3.74 μm SLM corresponds to over 120 mm with typical 8 μm 1080p SLM under the same sampling condition (rationale based on sampling analysis is provided in Supplementary Note 3).

In addition to the rationale behind the selected working range, we would also like to highlight that SFO-solver demonstrates strong generalization beyond the 85–115 mm training interval. As detailed in Results 2.3 of the revised manuscript and in Supplementary Note 6, we conducted a depth extension experiment to evaluate the model's performance outside the training range. The results show that SFO-solver maintains stable reconstruction quality—achieving PSNR above 30 dB—even across an extended range of 80–120 mm. This behavior supports the notion that our network not only captures the correct physical priors during training, but also generalizes well to unseen distances, underscoring the robustness and flexibility of our physics-constrained design.

Reviewer #1 Point 1.5 The manuscript lacks crucial information regarding the dataset used for training the SFO. Regarding the dataset, the authors only mention using the DIV2K dataset, it would be highly beneficial for the readers if the authors could showcase samples from the training set. Visualizing the input intensity images and their corresponding distances would help the reviewers. How about the test performance to other nature image dataset? Please provide more test details for dataset diversity performance.

Reply: Reviewer #1 kindly requested clarification about the dataset used for training and testing, and whether the testset is diverse enough. We agree that this point is important for clarity and have revised the manuscript accordingly. We confirm that our training images are only from DIV2K trainset (number: 0001-0800) while distances are randomly sampled between 85 mm and 115 mm to form the input data pairs, as illustrated in Supplementary Fig. S5. We respectfully note that the natural images in DIV2K dataset includes sufficient frequency components to allow our SFO-solver to generalize well. Our testset already consists of diverse image styles, including natural objects, animation frames and structured emblems. In the revised manuscript Methods 4.2 and Supplementary Note 4, we include visualizations of training samples from DIV2K and representing test samples to demonstrate coverage and variety. Reviewers can access the test results following the instructions in <https://github.com/NeoLiu02/CGH-SFO-solver> .

Reviewer #2 Point 2.1: The authors claim that one advantage of SFO-solver is that it can perform fast CGH calculations. However, I am quite skeptical about this claim. The authors showed that a 4K CGH can be computed in 0.16s. The GPU used was v100. I computed a 4K CGH on a GPU (RTX4060Ti) as a trial, and the computation time was almost the same as 0.16s. The GPU I used performs worse than the v100. Furthermore, it should be noted that my calculations were done with zero padding, so the actual calculations were done at 8K. At propagation distances where zero padding is not required, SFO-Solver may be considerably slower than a typical calculation.

Reply: Reviewer #2 raised valuable concerns regarding the inference speed of SFO-solver and its comparative performance relative to other CGH algorithms. We appreciate the reviewer's careful examination of our performance claims. To better address the reviewer's questions, we would like to first clarify the broader context of CGH algorithm development before addressing the specific concern.

As stated in the first paragraph of the revised manuscript Introduction Section, accelerating CGH computation has been a central motivation for the adoption of deep learning-based approaches since around 2020. Traditional non-learning methods (with the exception of DPAC, which is discussed in detail below) are inherently constrained by a trade-off between image quality and computational runtime. This landscape was clearly outlined by Peng et al. in the widely-recognized “Neural Holography”² paper, and has been commonly acknowledged in the CGH research community (refer to Fig. R2, note that all images in this work are in 1080p resolution). Similar comparison is also listed in our Supplementary Note 5.

[editorial note: figure redacted]

Fig. R2. Algorithm performance overview reported in “Neural Holography”²

Reviewer #2 mentioned conducting a trial to compute a 4K CGH using an RTX 4060 Ti GPU, reporting a computation time comparable to our stated 0.16 seconds. We appreciate this initiative and would like to provide some additional context. If a traditional non-learning algorithm (e.g., GS or a similar iterative method) was used in the trial, the result is likely subject to the inherent trade-off between speed and quality. Based on our experience, achieving a 0.16 s runtime with such methods typically results in simulated reconstructions with PSNR below 25 dB, accompanied by visible artifacts and increased speckle noise. In contrast, SFO-solver achieves remarkably high-quality reconstructions—exceeding 39 dB PSNR—within the same time budget, effectively breaking the speed–quality trade-off while maintaining the flexibility often lacking in other learning-based CGH networks.

We would also like to correct a point in the reviewer’s comment regarding hardware performance. As shown in Table R2, the RTX 4060 Ti (FP16 \approx 353 TFLOPS, FP32 \approx 22 TFLOPS) used by the reviewer offers higher raw tensor throughput than the NVIDIA V100 (FP16 \approx 118.5 TFLOPS, FP32 \approx 14.8 TFLOPS) used in our experiments. Therefore, under identical algorithmic settings, the same computation would likely take longer on our system.

Table R2. GPU performance reference table

Specification	Quadro GV100 (Volta)	RTX 4060 Ti (Ada Lovelace)
Release Year	2018	2023
Architecture	Volta	Ada Lovelace
CUDA Cores	5,120	4,352
Base / Boost Clock	1,130 MHz / 1,445 MHz	2,310 MHz / 2,535 MHz
VRAM	32 GB HBM2	8 GB / 16 GB GDDR6
Memory Bandwidth	870 GB/s	288 GB/s
Memory Bus Width	4096-bit	128-bit
Tensor Cores	640	136
RT Cores	N/A	34 (Ray Tracing cores)
FP32 Performance	~14.8 TFLOPS	~22 TFLOPS
Tensor Performance	~118.5 TFLOPS (FP16 Tensor)	~353 TFLOPS (FP16 Tensor)
Ray Tracing Performance	N/A	~51.1 RT TFLOPS
Power Consumption (TDP)	250 W	160 W

The reviewer also raised a point about zero padding. We would like to clarify that zero padding is indeed applied during forward optical diffraction within our self-supervised training loop. Therefore, the precision achieved by SFO-solver is consistent with that of the reviewer's 8K trial. For "propagation distances where zero padding is not required", we can also reduce the sampling density for SFO-solver to enable faster model inference. The overarching conclusion from these benchmark comparisons is that, under equivalent computational conditions (including resolution, padding, and GPU hardware, etc.), SFO-solver delivers significantly higher image quality within a short inference time compared to non-learning methods. Furthermore, unlike existing learning-based approaches, SFO-solver offers valuable flexibility in handling varying propagation distances—an essential capability for practical holographic applications that traditional CGH networks do not support. We have also included a paragraph discussing the positioning of our SFO-solver within the CGH research community. For a more thorough discussion, we respectfully refer the reviewer to the last paragraph of our *response to Reviewer #1, Point 1.1*.

Reviewer #2 Point 2.2: Of course, I understand that SFO-solver can achieve high image quality. However, the DPAC shown in Fig S3, for example, shows a reasonably high image quality of 25 dB, and for large propagation distances, the difference in image quality between SFO-solver and DPAC would be even smaller. In what computing environment were Fig S3's GS, DPAC and SDG run - SDG was probably run on a GPU, but what about GS and DPAC? (Reply: All GPU) In common with the concerns mentioned earlier, the computation time for DPAC seems to be slower.

Reply: Reviewer #2 also raised questions about the DPAC algorithm and its benchmarking. First, we sincerely appreciate the reviewer's observation regarding a time calculation oversight in our original benchmark trials. Specifically, GPU initialization time was not accounted for when measuring the performance of non-learning algorithms. This omission inadvertently led to an underestimation of their runtime. We have since corrected this in **the revised version of Supplementary Note 5**, ensuring all runtimes now reflect neat computation time without GPU initialization. For transparency and reproducibility, we have also uploaded the source code used for these comparisons at <https://github.com/NeoLiu02/CGH-SFO-solver>. Second, reviewer #2 mentioned that

DPAC seems to be a good algorithm that achieves both acceptable image quality and fast computation. This is indeed an insightful observation. Metric-wise we agree with the reviewer's statement, however, DPAC tends to underperform in experimental settings. This is primarily due to several practical limitations, including its inherently low optical efficiency, susceptibility to phase wrapping artifacts, and challenges in aligning filtering during optical reconstruction. These factors significantly degrade the quality of real-world holographic reconstructions, even if simulation results appear acceptable. We refer the reviewer to *Supplementary Note S6 of Ref. [3]* for a comprehensive analysis of these issues.

Reviewer #2 Point 2.3: What type of holographic display is SFO-solver targeting - is it a Near-eye display?

Reply: Reviewer #2 kindly inquired whether SFO-solver targets at holographic near-eye display. This is a valuable question for understanding the application context of SFO-solver. Currently our prototype is indeed a typical holographic near-eye display setup. We have demonstrated that now with SFO-solver, the depth position of the displayed objects can be adjusted with high quality, flexibility, and low latency. We believe this is a crucial technique to advance holographic applications in a more user-specific aspect. Also, because SFO-solver can be generally treated as an efficient neural solver for the general optimization problem in Eq. (1), we believe it has more potential in fields like optical lithography and metasurface design.

Reviewer #2 Point 2.4: The current SFO-solver can produce CGH with a propagation distance of 85 mm to 115 mm. Can this range be extended further? If it cannot be widened, the reasons for this should be discussed.

Reply: Both reviewers raised questions about why the working range of 85–115 mm was selected and whether it is sufficient. We acknowledge that a clear explanation of the experimental parameters is important for readers to comprehend our manuscript and sincerely thank the reviewers for offering these valuable suggestions. We addressed this

in the second paragraph of the revised manuscript Discussion section, noting that the central distance of 100 mm is consistent with prior literature, where the spatial footprint of optical elements commonly dictates working distances in the range of 100–200 mm. The 30 mm depth range provides sufficient perceptual defocus cues when using a 4K SLM with 3.74 μm pixel pitch, as verified in our optical experiments. Also, it's important to note that the 30 mm depth range that we choose with 3.74 μm SLM corresponds to over 120 mm with typical 8 μm 1080p SLM under the same sampling condition (rationale based on sampling analysis is provided in Supplementary Note 3).

Reviewer #2 additionally asked whether the depth range could be extended and what limits this flexibility. This is indeed an important point for understanding the method's physics interpretability. We conducted depth extension experiments (detailed in the revised second paragraph of the manuscript Results 2.3 and Supplementary Note 6, Fig. S7 and S9) to evaluate performance on distances beyond the training range. These results show that our model generalizes well over a 40 mm depth span (PSNR approximately over 30 dB). It also provides strong evidence that SFO-solver indeed effectively learns the underlying physics of optical wave propagation because for unseen distances SFO-solver still achieves relatively good results. For broader distance range the sampling density needs to be increased according to our sampling analysis.

Reviewer #2 Point 2.5: The current SFO-solver requires computation time proportional to the number of layers to perform layer hologram-like calculations. This seems to sacrifice computation time.

Reply: We thank Reviewer #2 for raising this point. Since the term "layer hologram-like calculations" could be interpreted in more than one way, we provide two responses corresponding to different interpretations of "layer".

First, if the reviewer is referring to "layers" as the neural network layers, we would like to clarify that the SFO-solver doesn't perform "layer hologram-like calculations" in the manner typical of convolution neural networks (CNNs) do. Unlike CNNs where deeper architectures are often required to progressively expand the receptive field, our model leverages spatial and Fourier operator (SFO) blocks that inherently provide global context.

The design of learnable Fourier filtering allows a single SFO block to achieve full receptive field coverage while encoding distance-dependent behavior. Therefore, we can address a 30 mm depth range using only six SFO blocks with a $3.74\ \mu\text{m}$ pixel pitch SLM. For comparison, the well-known “Tensor Holography”⁶ by Shi et al. (2021) uses 30 convolutional layers to address just a 6-mm optical path length under an $8\ \mu\text{m}$ SLM. In our case, increasing the depth range simply involves denser sampling in the learned Fourier filter—leading to higher memory use but little increase in computation time because we don’t stack sequential convolution layers to perform “hologram-like calculations”. We have updated the text in the revised first paragraph of the manuscript Results 2.2 and Supplementary Note 1 to reflect this clarification.

Second, if the reviewer is referring to “layers” as in the context of the layer-based method we used for synthesizing 3D holograms, we would like to clarify that although each depth layer of the 3D scene needs to be processed individually by SFO-solver, this computation can be efficiently parallelized. As described in revised manuscript Results 2.5, multiple input pairs (target intensity and propagation distance) can be packed into a single batch for simultaneous inference. This batching strategy ensures that the generation of multi-plane 3D CGHs introduces minimal additional computation time.

Reviewer #2 Point 2.6: The SFO-solver can freely shift the propagation distance. On the other hand, the authors show color reproduction results, but can the SFO-Solver also freely set the wavelength? Or did the SFO-Solver with color reproduction train three neural nets corresponding to RGB?

Reply: Reviewer #2 kindly inquired whether SFO-solver could be extended to handle different wavelengths. This is an interesting point regarding the model's adaptability. In the third paragraph of the revised manuscript Discussion Section, we explain that SFO-solver can in principle encode other physical variables, such as wavelength, due to the shared circular symmetry of the AS filter. However, we didn’t demonstrate it in our work because application-wise, a 3-channel laser is the typical setup to achieve color holographic display. In this case, separately training three models would do the work, and the flexibility in the wavelength is not desired.

To Reviewer #2

Reviewer #2 Point 2.7: Writing issue and usage of abbreviations.

The wording and language flow have been improved for every sentence in the manuscript, and use of abbreviations has been reduced to minimum. We believe the readability has been improved a lot after revision.

Summary table for revisions

Summary for revisions in the manuscript and the supplementary material

Reviewers' comments by points	Revisions regarding to the comments
1. SFO-solver's network structure #1: Impact of circular constraint and Fourier embedding #2: Number of layers	#1: Justification of circular constraint through model parameter counts (see Manuscript Results 2.2 & Supplementary Note 2); Ablation studies for Fourier embedding (see Manuscript Methods 4.1 & Supplementary Note 6) #2: Illustrated in our Detailed Feedback for Reviewers: #2 2.5
2. The depth range of 85-115mm #1 #2: Choice of this depth range #2: Extension of depth range	#1 #2: Explanation of 85-115 mm working distance range selection (see Manuscript Discussions & Supplementary Note 1); #2: Depth range extension test of SFO-solver (see Manuscript Results 2.3 & Supplementary Note 6)
3. Training and testing condition #1: Trainset visualization and concern on testset diversity	#1: Visualization of trainset and multi-styled testset images (see Manuscript Methods 4.2 & Supplementary Note 4)
4. Further applicability #1: Optical demonstration on more complex 3D scenes #2.1: Possibility of SFO-solver's wavelength-adaptivity #2.2: Application scenario	#1: Additional experiments on more complex 3D scenes (see Manuscript Results 2.5 & Supplementary Note 9) #2.1: Further discussion on wavelength-adaptivity (see Manuscript Discussion) #2.2: Illustrated in our Detailed Feedback for Reviewers #2 2.4
5. Benchmark comparisons #1: Need to compare with "Conditional Neural Holography" #2.1: Questions regarding SFO-solver's inference speed and the computing environment #2.2: Questions regarding DPAC	#1: Illustrations of "Conditional Neural Holography" in our Detailed Feedback for Reviewers: #1 1.1 (also see Manuscript Discussion) #2.1: Crucial background statements in Manuscript Introduction, and revised comparing results in Supplementary Note 5 (with further explanations in our Detailed Feedback for Reviewers: #2 2.1) #2.2: Illustrated in our Detailed Feedback for Reviewers: #2 2.2

Summary table for revisions

6. Manuscript's writing #2: Use of abbreviations	#2: The wording and language flow have been improved for every sentence in the manuscript, and use of abbreviations has been reduced to minimum.
--	---

Reference

1. Asano, Y., Yamamoto, K., Fushimi, T. & Ochiai, Y. Conditional neural holography: a distance-adaptive CGH generator. *Opt. Express* **33**, 16671-16683 (2025).
2. Peng, Y., Choi, S., Padmanaban, N., Kim, J. & Wetzstein, G. Neural Holography. In *ACM SIGGRAPH 2020 Emerging Technologies*. Article 8 (Association for Computing Machinery).
3. Peng, Y., Choi, S., Padmanaban, N. & Wetzstein, G. Neural holography with camera-in-the-loop training. *ACM Trans. Graph.* **39**, Article 185 (2020).
4. Maimone, A., Georgiou, A. & Kollin, J. S. Holographic near-eye displays for virtual and augmented reality. *ACM Trans. Graph.* **36**, Article 85 (2017).
5. Wu, J., Liu, K., Sui, X. & Cao, L. High-speed computer-generated holography using an autoencoder-based deep neural network. *Opt. Lett.* **46**, 2908-2911 (2021).
6. Shi, L., Li, B., Kim, C., Kellnhofer, P. & Matusik, W. Towards real-time photorealistic 3D holography with deep neural networks. *Nature* **591**, 234-239 (2021).